# Thermodynamic and Kinetic Sequence Selection in Enzyme-Free Polymer Self-Assembly inside a Non-equilibrium RNA Reactor

**DOI:** 10.3390/life12040567

**Published:** 2022-04-10

**Authors:** Tobias Göppel, Joachim H. Rosenberger, Bernhard Altaner, Ulrich Gerland

**Affiliations:** Physics of Complex Biosystems, Technical University of Munich, 85748 Garching, Germany; tobias.goeppel@tum.de (T.G.); joachim.h.rosenberger@tum.de (J.H.R.); bernhard.altaner@tum.de (B.A.)

**Keywords:** emergence of life, templated ligation, enzyme-free self-assembly, informational polymers, prebiotic evolution, enzyme-free replication, RNA reactor, autocatalytic set

## Abstract

The RNA world is one of the principal hypotheses to explain the emergence of living systems on the prebiotic Earth. It posits that RNA oligonucleotides acted as both carriers of information as well as catalytic molecules, promoting their own replication. However, it does not explain the origin of the catalytic RNA molecules. How could the transition from a pre-RNA to an RNA world occur? A starting point to answer this question is to analyze the dynamics in sequence space on the lowest level, where mononucleotide and short oligonucleotides come together and collectively evolve into larger molecules. To this end, we study the sequence-dependent self-assembly of polymers from a random initial pool of short building blocks via templated ligation. Templated ligation requires two strands that are hybridized adjacently on a third strand. The thermodynamic stability of such a configuration crucially depends on the sequence context and, therefore, significantly influences the ligation probability. However, the sequence context also has a kinetic effect, since non-complementary nucleotide pairs in the vicinity of the ligation site stall the ligation reaction. These sequence-dependent thermodynamic and kinetic effects are explicitly included in our stochastic model. Using this model, we investigate the system-level dynamics inside a non-equilibrium ‘RNA reactor’ enabling a fast chemical activation of the termini of interacting oligomers. Moreover, the RNA reactor subjects the oligomer pool to periodic temperature changes inducing the reshuffling of the system. The binding stability of strands typically grows with the number of complementary nucleotides forming the hybridization site. While shorter strands unbind spontaneously during the cold phase, larger complexes only disassemble during the temperature peaks. Inside the RNA reactor, strand growth is balanced by cleavage via hydrolysis, such that the oligomer pool eventually reaches a non-equilibrium stationary state characterized by its length and sequence distribution. How do motif-dependent energy and stalling parameters affect the sequence composition of the pool of long strands? As a critical factor for self-enhancing sequence selection, we identify kinetic stalling due to non-complementary base pairs at the ligation site. Kinetic stalling enables cascades of self-amplification that result in a strong reduction of occupied states in sequence space. Moreover, we discuss the significance of the symmetry breaking for the transition from a pre-RNA to an RNA world.

## 1. Introduction

Extant biological systems use different molecules for the storage of genetic information than for the catalysis of biomolecular reactions. Inevitably, the question arises of which came first, the informational polymer carrying the instructions for the enzymes, or the enzymes assembling the polymers? As RNA cannot only store genetic information but also fold into catalytically active structures [1,2,3,4,5], it is central to one of the most prominent hypotheses for the emergence of living systems [6,7,8,9,10,11]. However, this *RNA world hypothesis* does not explain the origin of the catalytic RNA molecules, called ribozymes [12]. While recent experimental work revealed potential prebiotic pathways to synthesize nucleotides [13,14,15], the mechanisms assembling these building blocks into functional molecules are only beginning to be explored [16,17,18,19,20,21]. The smallest ribozymes known today are 30 to 100 nucleotides long [22,23,24]. More complex ribozymes that could, e.g., assist replication are likely to have a minimum length of more than 150 nucleotides [25,26,27]. For polymers of a length between 30 to 150, a total of $10^{18}$ to $10^{90}$ distinct sequences are possible. However, the subset of catalytically active sequences is generally believed to be tiny. Hence, the spontaneous emergence (and maintenance) of a functional ribozyme in a random pool of oligonucleotides seems highly unlikely. Therefore, it remains unclear how the transition from a pre-RNA world lacking ribozymes to an RNA world where essential processes are driven by ribozymes could occur.

To understand this transition, one must study the dynamics in sequence space, which emerges when mononucleotides and short oligonucleotides inside a reaction volume collectively self-assemble into longer strands [28,29]. The self-assembly is governed by the process of *templated ligation* [30,31,32,33,34]. In this process, two strands that are hybridized adjacently on a third template strand are covalently linked. In contrast to random ligation concatenating two arbitrary strands, the process of templated ligation is sequence-selective for two reasons, a thermodynamic and a kinetic one. First, complementary nucleotide pairs at the hybridization sites increase the stability of the complex of strands, and thereby also increase the probability for a new covalent bond to be formed. However, complementary sequences of the same length comprising different sequence motifs can have different binding stabilities. The stability of a complex is determined by the hydrogen bonding between complementary base pairs and the stacking interactions between neighboring base pairs [35]. Changing the order of the base pairs or flipping one of the pairs generally alters the complex’s stability. Hence, certain sequence motifs can be favored over others thermodynamically [35,36,37]. Second, the kinetics of the ligation step are motif-selective: Non-complementary nucleotide pairs in the vicinity of the ligation site stall the formation of a new covalent bond. As a result, the formation of new strands from shorter fragments that do not match the template strand is also suppressed kinetically [32,38,39,40].

Experimentally probing the enzyme-free self-assembly of long strands from a random pool of mononucleotides and short fragments is challenging. Typically, the experiments require long times, while the reaction yields remain low and undesired side products obscure the results. Moreover, tracking the evolution of the whole sequence pool simultaneously remains an unsolved technical challenge [20,21,41,42]. Due to these constraints, non-enzymatic self-assembly experiments either employed initial oligonucleotides with precisely designed sequences limiting the product space [43,44,45,46,47] or focused on *primer extension* scenarios. In the latter scenario, a defined primer that is statically bound to a defined longer template strand gets extended by mononucleotides and short oligomers [38,48,49,50,51,52,53,54]. Two explorative experimental studies investigated the emergence of progressively longer strands from DNA-oligomers [18,55], both using DNA ligases to accelerate the assembly dynamics and to obtain better yields, and employing temperature cycling for strand separation. In Ref. [55], all possible 12-mers that can be formed from a binary alphabet of A and T are present initially. The assembly dynamics give rise to structured sequence pools characterized by a reduced sequence entropy compared to a random pool. The emerging longer strands are either characterized by a large A or T content since mixed strands are more prone to self-inhibition due to hairpin formation. In Ref. [18], three pairs of carefully designed complementary sequences composed of 20 nucleotides were used as basic building blocks. The authors demonstrated that certain subsets of sequence motifs composed of two basic building blocks form cooperative networks. Since the initial building blocks are already quite long in both studies, the binding energies of bound strands are large, such that small differences in the stacking energies associated with adjacent nucleotide pairs become irrelevant. However, subtle differences in the stacking energies might trigger sequence selection already on the level of the shortest oligomers, i.e., dimers and trimers, for which dissociation occurs spontaneously, rather than being induced externally. Since a sequence bias emerging early on might feedback onto itself, it could have a substantial impact on the pool of longer strands at later stages. In summary, an experimental study exploring growth dynamics into longer polymers starting from a pool of small building blocks is still missing.

Investigating the collective growth from small building blocks theoretically or by means of computer simulations in a model including the essential features of self-assembly, i.e., sequence-dependent (de)hybridization and ligation dynamics, is also challenging: First, the number of possible complex configurations grows exponentially fast as strands become longer. Second, there is an intrinsic separation of time scales between the fast dissociation of short and the slow dissociation of long hybridization sites and the slow ligation step. To date, no theoretical study on self-assembly via templated ligation accounted for the motif-dependent thermodynamic and kinetic aspects of hybridization and bond formation (see Section 4.2). Therefore, the following questions remained open: (1) What are the emerging dynamics in sequence space as strands grow longer? (2) Which are critical factors that enable self-enhancing sequence selection? (3) How do motif-dependent thermodynamic and kinetic parameters affect the selection process?

In Ref. [56], we developed a simulation method that partially handled the complexity of the self-assembly process. In this first study, we treated the sequence dependence of the (de)hybridization dynamics in a mean-field picture, in which the dissociation rate only depends on the length of the hybridization site. Our study identified several growth regimes arising from the competition of timescales for dissociation and extension. Moreover, we showed that, depending on external control parameters, the strand length distribution in the stationary state can exhibit a non-monotonous shape characterized by a distinct strand length. For the present study, we extended the simulation method to explicitly treat sequences, including sequence-dependent thermodynamics and kinetics. The ‘RNA reactor’ simulations that we report here assume a closed reaction volume, initialized with mononucleotides and a few dinucleotides, with an unbiased nucleotide distribution (symmetric initial condition in sequence space). Within the RNA reactor, oligomers grow via templated ligation and degrade via hydrolysis. Eventually, the sequence pool converges to a non-equilibrium stationary state characterized by its length and sequence distribution.

To address the above questions, we consider different model variants. We start with a simple reference scenario, where kinetic stalling is absent and the stacking energies for all complementary neighboring nucleotide pairs are identical. This scenario distinguishes solely between complementary and non-complementary pairings. We then introduce thermodynamic and kinetic sequence selection, both separately and in combination, and compare the resulting four different scenarios. Our main finding is that, under the conditions assumed here, thermodynamic discrimination within hybridized strands is not sufficient by itself to promote self-enhanced sequence selection that drives the sequence pool significantly away from the random state. However, distinct patterns in sequence space arise if non-complementary strand termini at the ligation site slow down the ligation step significantly (kinetic stalling). In this case, a small thermodynamic bias for certain sequence-motifs triggers a self-enhancing dynamics, such that the thermodynamically favored sequence motif dominates the stationary state.

## 2. Models and Methods

### 2.1. Strands and Complexes

We consider a binary alphabet composed of two complementary nucleotides, denoted as *X* and *Y* for generality. A molecule containing *L* nucleotides linked covalently is called a *strand* of length *L* (see Figure 1a). A single nucleotide is a strand of L=1. Strands are directed and point from the 5${}^{\prime }$ to the 3${}^{\prime }$ end, which we also refer to as the − and the + ends. We allow strands to hybridize to each other, but do not account for the possibility of self-folding. An entity formed by several hybridized strands is referred to as a *complex*. All staggered conformations that can arise from a set of single strands are allowed inside the RNA reactor, regardless of the number of strands and *mismatches*, i.e., non-complementary nucleotide pairs (see Figure 1b,c and Figure A1 in Appendix B). However, branched hybridization structures and other nonlinear complexes involving loops are excluded.

We call a complex that contains two or three strands a *duplex* or *triplex*, respectively. The overlapping horizontal region between two strands is referred to as a *hybridization site*. Moreover, the vertical interface between two strands hybridized adjacently on a third strand is called a *ligation site*.

**Figure 1 life-12-00567-f001:**
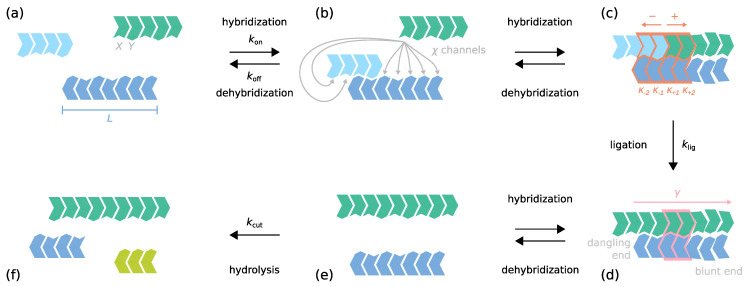
Schematic illustration of the dynamics inside the RNA reactor. The elementary processes are hybridization, dehybridization, ligation on the template, and hydrolysis with corresponding elementary rates $k_{\mathrm{on}}$, $k_{\mathrm{off}}$ and $k_{\mathrm{lig}}$, and $k_{\mathrm{cut}}$. The elementary rates $k_{\mathrm{on}}$, $k_{\mathrm{off}}$, $k_{\mathrm{lig}}$ are functions of the sequence context: (**a**) Strands have a binary sequence and are directed. *L* denotes their length. (**b**) When two molecules collide, they can form χ different hybridization complexes. (**c**) Hybridization sites within complexes (horizontal interfaces) can contain mismatches. Two strands (− and + strand) located adjacently on another strand may get joined covalently via templated ligation. The speed of the ligation reaction depends on the complementarity κ of the nucleotide pairs at the ±1 and ±2 position (red box). Non-complementary pairings lead to kinetic stalling. (**d**) The stability of a hybridization site is governed by the hybridization energy $\Delta G_{\mathrm{hyb}}$. $\Delta G_{\mathrm{hyb}}$ is obtained by summing over stacking energies γ associated with nearest-neighbor blocks (purple box) and considering terminal nucleotide pairs. $\Delta G_{\mathrm{hyb}}$ and γ depend on the structural and sequential context. Mismatches weaken the binding. (**e**,**f**) Covalent bonds within single strands or single-stranded segments may get cleaved via hydrolysis at a constant rate. The resulting unactivated strand termini are assumed to be rapidly reactivated.

### 2.2. Elementary Reactions

Strands and complexes form new complexes via *hybridization*, *dehybridization*, *templated ligation* and *hydrolysis* (see Figure 1). All reactions are assumed to be elementary and occur with sequence- and structure-dependent rates $k_{\mathrm{on}}$, $k_{\mathrm{off}}$, $k_{\mathrm{lig}}$, and $k_{\mathrm{cut}}$. Assuming constant environmental conditions, $k_{\mathrm{on}}$ and $k_{\mathrm{off}}$ are related to the *hybridization energy*
$\Delta G_{\mathrm{hyb}}$ associated with a hybridization site via the thermodynamic consistency requirement [57]
(1)$$\frac{k_{\mathrm{off}}}{k_{\mathrm{on}}}=VN_{A}c_{\circ }\;e^{\beta \Delta G_{\mathrm{hyb}}}\;,$$
where $\beta ={\left(k_{B}T\right)}^{-1}$, $k_{B}$ is Boltzmann’s constant, and *T* denotes the (absolute) temperature, *V* and $N_{A}$ are the reaction volume and Avogadro constant and $c_{\circ }=1\;\mathrm{mol}/L$ is the reference concentration. We will express all concentrations as a multiple of the reference concentration. Moreover, in the following, we use the dimensionless hybridization energy
(2)$$\Gamma =\beta \Delta G_{\mathrm{hyb}}.$$
Γ is obtained by summing over dimensionless motif-dependent stacking energies of nearest-neighbor blocks [35,36,37] (see Section 2.4). Γ thus reflects the number of complementary and non-complementary nucleotide pairs and their arrangement. Generally, mismatches increase the hybridization energy, therefore reducing the stability of a complex.

The formation of new covalent bonds requires energy, which needs to be provided by the environment in the form of an activation chemistry [42,58,59]. We assume that the RNA reactor is constantly fueled with the activation chemistry, enabling a fast chemical (re)activation of the termini of all strands present in the system. With that, two strands that are located next to each other on a third strand can always ligate. The fast activation step is not modeled explicitly. The rate $k_{\mathrm{lig}}$ at which two neighboring strands ligate depends on the paired nucleotides in the vicinity of the ligation site. Mismatches lead to *kinetic stalling*, i.e., a reduction of the ligation speed [32,38,39,40].

We model the kinetic stalling using the *kinetic stalling factors* $\Phi_{\pm }\leq 1$. The stalling factors $\Phi_{\pm }$ are functions of the *complementarities*
$\kappa_{\pm i}\in \left\{1,0\right\}$ of the paired nucleotides in the vicinity of the ligation site. The value 1 indicates a complementary pair, whereas the value 0 indicates a non-complementary pair of nucleotides. $\Phi_{-}$ takes the complementarities $\kappa_{-1},\kappa_{-2}$ of the two nucleotides in the − direction of the ligation site into account, while $\Phi_{+}$ is an equivalent expression for the two nucleotides in the + direction (see Figure 1c,d and Section 2.5 for more details). The two stalling factors are then multiplied with the *basal ligation rate*
λ. With that, the ligation rate $k_{\mathrm{lig}}$ becomes
(3)$$k_{\mathrm{lig}}=\lambda \;\Phi_{-}\left(\kappa_{-1},\kappa_{-2}\right)\;\Phi_{+}\left(\kappa_{+1},\kappa_{+2}\right).$$ Since random ligation of two strands in the absence of a template is weak compared to templated ligation [31,32,33,34], we neglect it in our model.

While covalent bonds within double-stranded parts of complexes are assumed to be stable against hydrolysis, bonds within single-stranded sections get cleaved [60,61,62,63] (see Figure 1e,f). The corresponding rate is assumed to be sequence-independent,
(4)$$k_{\mathrm{cut}}=\mathrm{const}.$$ with that, the overall degradation rate for a single strand of length *L* is $\left(L-1\right)k_{\mathrm{cut}}$, for example. In real systems, $k_{\mathrm{cut}}$ varies by several orders of magnitude as a function of environmental parameters and crucially depends on the polymer’s backbone chemistry [60,62,63,64,65]. Note that templated ligation and cleavage are irreversible, since the respective reverse reactions (random ligation and “templated cleavage”) are absent in our model.

### 2.3. Kinetics of Hybridization and Dehybridization

Since Equation (Equation 1) only constrains the ratio of $k_{\mathrm{on}}$ and $k_{\mathrm{off}}$, an additional kinetic parameter is required to fix the kinetics of the model. However, the chosen parametrization has only a minor effect on the global kinetics, given that ligation and hydrolysis are rare compared to hybridization and dehybridization (see Section 2.8). Our approach uses a constant rate of collision between two complexes $k_{\mathrm{coll}}={\left(VN_{A}c_{\circ }t_{0}\right)}^{-1}$, where $t_{0}$ is the collision time scale. In the following, we express all times in units of collision time scale $t_{0}$.

In general, two colliding complexes can form multiple hybridization configurations via χ distinct hybridization channels (see Figure 1b). We assume no bias for any channel, such that the probability of choosing one particular channel is
(5)$$p_{\mathrm{hyb}}=1/\chi .$$ Hence, the rate for a hybridization via a given channel is
(6)$$k_{\mathrm{on}}=k_{\mathrm{coll}}\;p_{\mathrm{hyb}},$$
whereas the dehybridization rate becomes
(7)$$k_{\mathrm{off}}=\frac{1}{\chi }e^{\Gamma }.$$ If χ=0, no hybridization can occur. This is the case if one of the colliding complexes is a duplex without any overhang. A parametrization attributing the hybridization energy Γ to the dehybridization rate is common in theoretical approaches and is consistent with experiments [66,67,68]. The kinetic model resulting from the specific choice of $k_{\mathrm{on}}$ and $k_{\mathrm{off}}$, i.e., Equations (Equation 6) and (Equation 7) was developed, described in detail, and rationalized in Ref. [56], where we studied self-assembly in a sequence-independent model with hybridization energies simply being proportional to the overlap lengths. The kinetic assumptions Equations (Equation 6) and (Equation 7) reduce the computational complexity considerably, while still sampling complexes in a thermodynamically consistent way (see Appendix C).

### 2.4. Hybridization Energy

Detailed models for the free energy of given RNA and DNA secondary structures [35,36,37] build on the so-called stacking interactions of neighboring nucleotide pairs at the hybridization sites [35,69]. Every nearest-neighbor interaction, i.e., every block of two adjacent nucleotide pairs, is associated with a motif-dependent stacking energy. These stacking energies additively contribute to the total free energy. Additional contributions to the total free energy take into account nonlinearities of secondary structures such as loops, branching points, and particular end configurations.

Our coarse-grained model that excludes nonlinear complex structures conserves the essential feature of the detailed nearest-neighbor models. The central element of our energy model is the stacking interaction of two neighboring nucleotides pairs $P_{i}$ and $P_{i+1}$ with
(8)PiandPi+1∈X·Y,Y·X,X·X,Y·Y,
where dots symbolize hydrogen bonds between complementary nucleotides. To every block of adjacent nucleotide pairs $\left[P_{i}P_{i+1}\right]$, we assign a dimensionless stacking energy $\gamma \left(\left[P_{i}\;P_{i+1}\right]\right)$. (Note that the last two pairs are non-complementary. Therefore, the nucleotides are not connected via a dot.) The hybridization energy is then given by the sum over all stacking energies and contributions $\epsilon_{-}$ and $\epsilon_{+}$ accounting for the terminal nucleotide pairs at the − and the + end of double-stranded segment (see Figure 1d), i.e.,
(9)$$\Gamma =\sum_{i\;\in \;\begin{array}{c} \mathrm{blocks} \end{array}}\gamma_{i}+\epsilon_{-}+\epsilon_{+}.$$ The contributions $\epsilon_{\mp }$ for the ∓ end also depend on the structural and sequence context. If the ∓ terminal nucleotide pair forms a *dangling end*, i.e., is preceded or followed by an unpaired nucleotide, we have $\epsilon_{\mp }\neq 0$. If the terminal nucleotide pair is part of a ligation site, there also is a contribution $\epsilon_{\mp }\neq 0$. If otherwise, it corresponds to *blunt end* of a complex, and we have $\epsilon_{\mp }=0$ (see [App app111-life-12-00567] for details).

For simplicity, we assume symmetric stacking energies, i.e., $\gamma \left(\left[P_{i}\;P_{i+1}\right]\right)=\gamma \left(\left[P_{i+1}\;P_{i}\right]\right)$. Moreover, complementary nearest-neighbor blocks are either *alternating* if
(10)PiPi+1∈X−·−Y−Y·X,Y−·−X−X·Y,
or *homogeneous* if
(11)PiPi+1∈X−·−Y−X·Y,Y−·−X−Y·X. Here, the − symbol stands for a covalent bond. We denote stacking energies assigned to alternating and homogeneous blocks by $\gamma_{\mathrm{alt}}$ and $\gamma_{\mathrm{hom}}$. Motivated by the observation that $\gamma_{\mathrm{alt}}\neq \gamma_{\mathrm{hom}}$ in DNA and RNA systems (see Table 1) [36,37], we treat the energy difference
(12)$$\Delta \gamma =\gamma_{\mathrm{alt}}-\gamma_{\mathrm{hom}}.$$
as a variable parameter. Without loss of generality, we assume Δγ≤0 for our model. Moreover, we assume constant stacking energies $\gamma_{1\mathrm{nc}}$ and $\gamma_{2\mathrm{nc}}$ for nearest neighbor blocks containing one or two non-complementary nucleotide pairs. Since blocks containing mismatches weaken the binding, their stacking contributions are positive. The contribution for a block with two mismatches is larger than for a block with only one mismatch. In summary, the block-wise contributions obey the hierarchy
(13)$$\gamma_{\mathrm{alt}}\leq \bar{\gamma_{\mathrm{com}}}\leq \gamma_{\mathrm{hom}}<0<\gamma_{1\mathrm{nc}}<\gamma_{2\mathrm{nc}},$$
where $\bar{\gamma_{\mathrm{com}}}$ is the average energy value of complementary blocks (see Table 2), i.e.,
(14)$$\bar{\gamma_{\mathrm{com}}}=\left(\gamma_{\mathrm{alt}}+\gamma_{\mathrm{hom}}\right)/2.$$

### 2.5. Kinetic Stalling

Our kinetic stalling model describes the experimentally observed sequence dependence [32,38,39] in a simplified way, using only two parameters, $\sigma_{1}$, $\sigma_{2}$. Mismatches directly at the ligation site affect the ligation speed more substantially than distant ones. If the nucleotide pair at the ±1 position is non-complementary ($\kappa_{\pm 1}=0$), a mismatch at the ±2 position ($\kappa_{\pm 2}=0$) amplifies the stalling effect. Otherwise, a mismatch at the ±2 position has no effect, i.e.,
(15)$$\Phi_{\pm }\left(\kappa_{\pm 1},\kappa_{\pm 2}\right)=\left\{\begin{array}{cc} 1 & \mathrm{for}\;\;\kappa_{\pm 1}=1\wedge \;\kappa_{\pm 2}\in \left\{0,1\right\}\\ \sigma_{1} & \mathrm{for}\;\;\kappa_{\pm 1}=0\;\wedge \;\kappa_{\pm 2}=1\\ \sigma_{1}\sigma_{2} & \mathrm{for}\;\;\kappa_{\pm 1}=0\;\wedge \;\kappa_{\pm 2}=0 \end{array}\right.,$$
where $\sigma_{1}\leq \sigma_{2}$. If the hybridization site in the + or − direction contains only one nucleotide pair (see Figure 1c), we use Equation (Equation 15) with $\kappa_{\pm 2}=1$.

The strength of the stalling effect depends on the underlying activation chemistry as well as the type of nucleotides being used [32,38,39]; therefore, we treat $\sigma_{1}$ and $\sigma_{2}$ as variable parameters (see Table 2).

### 2.6. Effective Cyclic Environment

According to the energy model defined in Equation (Equation 9), hybridization energies for long, primarily complementary hybridization sites become arbitrarily negative. Hence, the corresponding dehybridization rates converge to zero exponentially. As a result, strands can be bound in duplexes without single-stranded overhangs over long times. This effect is called template inhibition and leads to freezing of the dynamics [41,61,70]. To overcome this problem, we assume cyclic variations of the physico-chemical conditions (temperature, pH, or salt concentrations) inside the RNA reactor such that all hybridized strands separate within the period time τ [19,71]. Aforesaid oscillatory conditions arise for example due to convection flows induced by temperature gradients or micro scale water cycles at a heated gas–liquid interface. Both scenarios arise naturally in rock fissure in the vicinity of hydrothermal vents [72,73,74,75,76]. They are modeled effectively by introducing a lower bound for the dehybridization rate [56,77], i.e., modifying Equation (Equation 7) such that
(16)$$k_{\mathrm{off}}=\max\left\{\frac{1}{\chi }e^{\Gamma },k_{\mathrm{low}}\right\},$$
where $\tau =k_{\mathrm{low}}^{-1}$. With that, the (dis)assembly dynamics of long complementary complexes do not obey the thermodynamic consistency requirement Equation (Equation 1) anymore. Nonetheless, the kinetics are still plausible: The constant collision rate is a reasonable approximation for a collision process with a diffusion coefficient decaying with length compensated by a cross section growing with length [56].

### 2.7. Validity of Our Model and Application to Primer Extension

In the Results section, we focus on self-assembly scenarios where all strands (apart from monomers) are equally important because there are no distinct template, primer, and substrate strands as in typical primer-extension situations. However, in Appendix F, we show that our modeling of the kinetic stalling and the (de)hybridization kinetics in combination leads to copying dynamics in primer-extension situations consistent with the experimental literature.

### 2.8. Parametrization of Rates

We can parametrize every rate constant $k_{*}$ introduced so far by a dimensionless length $l_{*}$ such that
(17)$$k_{*}=e^{\bar{\gamma_{\mathrm{com}}}\;l_{*}}.$$ This presentation will prove convenient in the later analysis of the results as it connects time scales to length scales. For example, $l_{\mathrm{low}}=7$ tells us that entirely complementary hybridization sites composed of more than seven nucleotides dissociate as quickly as altogether complementary hybridization sites comprising exactly seven nucleotides. Parameters used in the following are summarized in Table 2. Moreover, $l_{\mathrm{lig}}=10$ signifies that the timescale of a dehybridization for a hybridization site counting more than ten nucleotides would be slower than the bare ligation timescale if the lower bound with $l_{\mathrm{low}}$ would not have been introduced.

### 2.9. Implementation

To simulate the model dynamics in C++, we use an extension of the framework developed in [56], based on an optimized Gillespie algorithm [78,79,80]. The simulation only keeps those species in memory that have a non-zero copy number. If a species appears (vanishes), the corresponding species object is created (deleted) dynamically.

## 3. Results

### 3.1. Boundary Conditions and Observables

We aim to investigate the model dynamics starting on the lowest level, i.e., where mononucleotide and a few short oligonucleotides collectively evolve into larger entities. Will the sequences of longer strands be random, or will they show patterns? We chose the arguably simplest setting for our study, which is a closed reaction volume that does not exchange complexes with the environment. In such a setting, we expect the dynamics to settle to a stationary state eventually. We initialize the reaction volume symmetrically with 5000 nucleotides distributed over 4920 mononucleotides and 40 dimers (see Figure 2a). Moreover, we adjust the reaction volume such that the total nucleotide concentration is given by $c_{\mathrm{tot}}=0.01\;c_{\circ }$. The ratio of the initial monomer to dimer concentration is $c_{1}^{\mathrm{init}}:c_{2}^{\mathrm{init}}=123:1$.

Our focus is on the evolution of the length distribution and the dynamics in sequence space. The length distribution $c_{L}$ expresses the concentration of strands of length *L*, irrespective of whether they are part of a complex or not. We denote the mean length by $\bar{L}$. To describe the dynamics in sequence space, we aim for a simple observable with an intuitive and straightforward interpretation. Therefore, we introduce the *zebraness* as a characterization of a strand’s sequence. The zebraness ζ(S) of a strand *S* of length $L_{S}$ is the number of alternating “zebra” submotifs X−Y or Y−X within its sequence divided by the number of binary motifs $L_{S}-1$ (see Figure 2b). With that, a random sequence $S_{r}$ is expected to have $\zeta (S_{r})=0.5$ on average. Moreover, the system-level zebraness *Z* characterizes how zebra-like the ensemble of strands is. It is given by
(18)$$Z=\frac{\sum_{S}\zeta \left(S\right)\;\left(L_{S}-1\right)}{\sum_{S}\left(L_{S}-1\right)},$$
where the summation is performed over all individual strands with L>1. A system containing homogeneous strands only would have Z=0, whereas, for a system exclusively composed of strands with alternating sequences, we would have Z=1. All plots show ensemble averages which are taken over 20 independent realizations of the dynamics.

### 3.2. Overview of Key Findings

Before we present the detailed analysis of the four different scenarios outlined in the introduction, we briefly summarize our key findings. First, we study the simplest variant of our model where both kinetic stalling and energetic bias are absent, i.e., Δγ=0 and $\sigma_{1}=\sigma_{2}=1$. This scenario only distinguishes between nearest-neighbor blocks containing zero, one, or two mismatches. Alternating and homogenous blocks have identical energetic properties. While non-complementary pairings decrease the complex’s stability, erroneous pairings at the ligation site do not reduce the bare ligation rate. The first model variant does not give rise to motif selection; the composition of the sequence pool remains entirely random, i.e., Z=0.5.

In the second scenario, we introduce an energetic bias Δγ<0 for alternating blocks while still assuming a non-discriminative ligation. The energetic bias favors the hybridization of strands with zebra-like sequences. This time, a weak zebra pattern with Z>0.5 is induced transiently during the initial growth phase. However, the pattern vanishes almost completely as the system approaches the steady-state (see Figure 3).

The dynamics in sequence space change drastically if kinetic stalling with $\sigma_{1},\sigma_{2}<1$ is applied. If non-complementary nucleotide pairs at the ligation site slow down the formation of a covalent bond, distinct patterns in sequence space can emerge. In the third scenario, we investigate the correlation between the strength of the kinetic stalling effect and the reduction of possible states in sequence space assuming identical energetic properties for alternating and homogeneous blocks, i.e., Δγ=0. Within this setting, we observe a spontaneous symmetry breaking in sequence space. Independent realizations of the dynamics evolve to stationary states, dominated by either zebra motifs with Z<0.5 or homogeneous motifs with Z>0.5 (see Figure 4). Moreover, we see that a dominant pattern emerges such that Z→0 or 1 if the stalling effect is strong enough.

In the fourth scenario, we show that a slight energetic bias Δγ<0 can become self-amplifying if kinetic stalling is present (see Figure 5). Depending on the strength of the kinetic stalling, the system converges to either a partial or pure zebra state characterized by either Z>0.5 or Z→1.

### 3.3. Reference Model without Energetic Bias and Kinetic Stalling

This section aims to answer whether the energetic discrimination of matches and mismatches alone is sufficient to give rise to spontaneous symmetry breaking in sequence space such that Z≠0.5. To this end, we study the simplest variant of our model with neither energetic bias nor kinetic stalling, i.e., Δγ=0 and $\sigma_{1}=\sigma_{2}=1$.

Initially, the growth dynamics of the mean length $\bar{L}$ is slow until $t\approx 8.8\times 10^{9}$ (see the dark blue curve in Figure 3a). At this point, the mean length $\bar{L}$ starts to increase rapidly. We refer to this time point as the *onset* of growth and denote it by $\hat{t}$. After the steep increase, $\bar{L}$ reaches a plateau value. The inset shows the steady-state length distribution displaying a double-exponential shape.

The ensemble average of the zebraness *Z* initially fluctuates and then converges to Z=0.5 (see Figure 3b). Looking at single trajectories (see Figure 3c) reveals a behavior similar to the ensemble average. The initial values of Z≶0.5 on the single trajectory-level are due to small numbers of strands with L>1. A value of Z=0.5 hints towards an entirely random sequence pool but does not exclude motif correlations on larger scales. However, analyzing distributions of longer motifs reveals that the final sequence composition is indeed random (see Appendix D).

**Figure 3 life-12-00567-f003:**
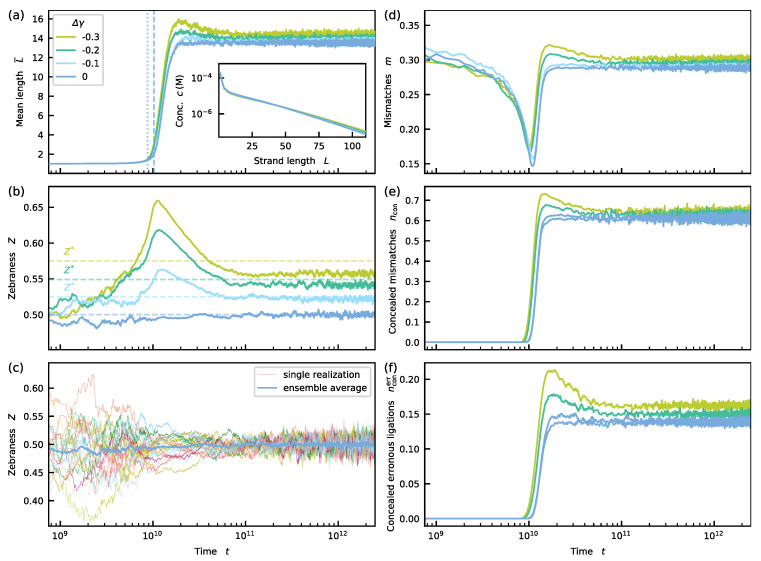
Mean length $\bar{L}$ and system-level zebraness *Z* as functions of time for $\sigma_{1}=\sigma_{2}=1$ and various Δγ. (**a**) A sharp increase of $\bar{L}$ appears at $\hat{t}$. For Δγ=0, the dashed line corresponds to $\hat{t}$ resulting from the formal definition, whereas the dotted line is the prediction obtained from Equation (Equation 19). For Δγ<0, $\bar{L}$ reaches a maximum before decaying gradually to the stationary value. The inset shows the steady-state length distributions. (**b**) If there is no energetic bias, i.e., Δγ=0, no distinct patterns emerge in sequence space, and hence Z=0.5 (see also Appendix D). If an energetic bias Δγ<0 is applied, *Z* grows initially and then decays when $\bar{L}\approx 7$. The final value is slightly above the random state Z=0.5 and below the simple thermodynamic estimate $Z^{*}$ (see Equation (Equation 22)). (**c**) Single realizations of the dynamics for Δγ=0 behave similar to ensemble average. Strong fluctuations for small times stem from low numbers of strands with L>1. (**d**) The fraction of mismatches *m* first decreases and then increases as the mean length becomes longer. (**e**) The fraction of concealed mismatches, i.e., mismatches not affected by energetic discrimination grows simultaneous with the mean length. (**f**) Over time, concealed erroneous ligations become frequent and destroy the initial sequence bias.

The evolution of the mean length shows some interesting features. After a lag phase, its increase becomes exponential at $t=\hat{t}$ (see Appendix A). Formally, we define the onset of growth $\hat{t}$ by intersecting the tangents to the $\bar{L}$–curve at t=0 and the point where the increase is strongest (dashed line in Figure 3a; for details, see Appendix E and Appendix A). We observe that $\hat{t}$ coincides with the moment in time at which *higher-order ligations*, i.e., ligations involving at most one monomer become more abundant than ligations joining two monomers to a dimer (see Figure A3 in Appendix E). Moreover, we can predict the onset with a relative error smaller than 15% by the following formula derived in Appendix E (dotted line in Figure 3).
(19)$$\hat{t}\approx \log\left[\frac{c_{\mathrm{tot}}^{2}k_{\left\{1,1|2\right\}}-k_{\mathrm{cut}}}{c_{\mathrm{tot}}c_{2}^{\mathrm{init}}k_{\left\{1,2|2\right\}}}\right]/\left(c_{\mathrm{tot}}^{2}k_{\left\{1,1|2\right\}}-k_{\mathrm{cut}}\right)$$ Here, $c_{2}^{\mathrm{init}}$ is the initial dimer concentration and $k_{\left\{1,1|2\right\}}$ and $k_{\left\{1,2|2\right\}}$ are the effective rate constants at which new dimers or trimers are formed from monomers or monomers and dimers. $k_{\left\{1,1|2\right\}}$ is given by
(20)$$k_{\left\{1,1|2\right\}}=\frac{k_{\mathrm{lig}}}{K_{\left\{1,1|2\right\}}},$$
where $K_{\left\{1,1|2\right\}}$ is the effective dissociation constant averaged over all triplexes involving two monomers and one dimer. An analogous expression exists for $k_{\left\{1,2|2\right\}}$. Repeating the computer experiment with $k_{\mathrm{cut}}$ and $k_{\mathrm{lig}}$ different from the standard values given in Table 2 confirmed the validity of Equation (Equation 19) (see Appendix A).

Moreover, altering $k_{\mathrm{cut}}$ and $k_{\mathrm{lig}}$ while keeping the other parameters fixed confirmed that the mean length $\bar{L}$ in the stationary state does not explicitly depend on these two variables but only on their ratio
(21)$$\bar{L}=\bar{L}\left(k_{\mathrm{lig}}/k_{\mathrm{cut}}\right)\;\;\mathrm{for}\;\;t\to \infty ,\;,$$
as expected from dimensional analysis. The dependence of the mean length on the ratio $k_{\mathrm{lig}}/k_{\mathrm{cut}}$ can be derived analytically for a random ligation model [81].

### 3.4. Energetic Bias in the Absence of Kinetic Stalling

In the previous section, we saw that the energetic discrimination between complementary and non-complementary nucleotide pairs alone is insufficient for the spontaneous emergence of patterns in sequence space. Therefore, we now ask whether an energetic bias Δγ<0 favoring the binding of zebra motifs can induce zebra patterns that become self-amplifying, while kinetic stalling is still absent ($\sigma_{1}=\sigma_{2}=1$).

The energetic bias, Δγ<0, causes a transient overshoot in the mean length beyond the steady-state value but otherwise does not strongly affect the dynamics of the mean length (Figure 3a). The onset of growth $\hat{t}$ can still be predicted by a formula analogous to Equation (Equation 19) (see Appendix E and Appendix A for plots of single realizations). The steep increase after the lag phase is followed by a gradual descent to the steady state value, slightly below the maximum. Moreover, the steady-state length distribution remains very similar to the scenario without energetic bias.

In sequence space, a simple thermodynamic estimate $Z^{*}$ for the final zebraness can be made based on a two-state system with an energy difference Δγ
(22)$$Z^{*}=\frac{1}{1+e^{\Delta \gamma }}.$$ Since this estimate neglects any correlation and feedback effects, one could naively expect that the zebranass resulting from the simulated model dynamics reaches a value larger than $Z^{*}$. Indeed, the observable initially grows beyond the estimate $Z^{*}$. However, the growth stops when the mean length reaches a value of $\bar{L}\approx 7$. At that point, the zebraness starts to decay and converges to a stationary value simultaneously with $\bar{L}$ (see Figure 3b). In the stationary state, the zebraness *Z* is only slightly above the result for random sequences, i.e., Z=0.5, and below the simple thermodynamic estimate $Z^{*}$.

### 3.5. Loss of Energetic Discrimination Prevents Sequence Selection

Why are the initially emerging zebra patterns triggered by the energetic bias Δγ<0 in Figure 3b neither amplified nor maintained? In the following, we analyze the growth processes in detail to give an intuitive explanation.

Strand growth requires the formation of complexes comprising at least three strands. The more negative the hybridization energies of the hybridization sites in these complexes are, the more stable the configurations become and the higher the probability for a ligation gets. Non-complementary nucleotide pairs increase the hybridization energy and, therefore, weaken the binding. To analyze the effect of these mismatches, we define the overall fraction of mismatches *m* as
(23)$$m=\frac{N_{\mathrm{non}}}{N_{\mathrm{pairs}}},$$
where $N_{\mathrm{pairs}}$ and $N_{\mathrm{non}}$ are the absolute numbers of nucleotide pairs and non- complementary nucleotide pairs in all present complexes. Initially, the fraction of mismatches *m* decreases since complexes mostly containing complementary nucleotides that emerge during the early growth persist longer and hence contribute more substantially to the average. However, the fraction of mismatches starts to increase when the mean length becomes larger (see Figure 3d). This increase of the mismatch fraction arises from the loss of thermodynamic discrimination induced by the cut-off $k_{\mathrm{low}}$ in the dehybridization rate, i.e., the effective temperature cycles (see Section 2.6). Although the hybridization energy may become arbitrarily negative for large hybridization sites, the dehybridization rate can not become smaller than the lower bound $k_{\mathrm{low}}$. The length scale associated with $k_{\mathrm{low}}$ is $l_{\mathrm{low}}=7$ (see Section 2.8). This implies that an entirely complementary hybridization site comprising more than seven pairs has the same stability as a mismatch-free hybridization site composed of exactly seven pairs. Moreover, mismatches in extended hybridization sites might have no effect on the rate for unbinding because the hybridization site still contains a high number of matches. If many matches are present, the hybridization energies are strongly negative such that the lower threshold still determines the rate for dehybridization. This effect enables *concealed mismatches*. Concealed mismatches are mismatches that do not increase the dehybridization rate $k_{\mathrm{off}}$ of a hybridization site. Replacing a concealed mismatch with a complementary pair would not decrease $k_{\mathrm{off}}$ further since it is already given by the cut-off, i.e., $k_{\mathrm{off}}=k_{\mathrm{low}}$. The longer the strands become during the first growth phase, the more concealed mismatches emerge. With the absolute numbers of mismatches and concealed mismatches in all present complexes $N_{\mathrm{non}}$ and $N_{\mathrm{con}}$, we now introduce the fraction of concealed mismatches $n_{\mathrm{con}}$ as
(24)$$n_{\mathrm{con}}=\frac{N_{\mathrm{con}}}{N_{\mathrm{non}}},$$ The evolution of $n_{\mathrm{con}}$ shown in Figure 3e reveals that most of the occurring mismatches are concealed, once $\bar{L}$ has become approximately twice as large as $l_{\mathrm{low}}$. Concealed mismatches also occur at the strand termini at ligation sites and may lead to the formation of new binary motifs which are not complementary to the templating motif at the ligation site. We call such a ligation involving at least one concealed mismatch a *concealed erroneous ligation*. Dividing the number of concealed erroneous ligations $N_{\mathrm{err}}$ per time by the overall number of ligations $N_{\mathrm{lig}}$ per time gives the fraction of concealed erroneous ligations $n_{\mathrm{con}}^{\mathrm{err}}$, i.e.,
(25)$$n_{\mathrm{con}}^{\mathrm{err}}=\frac{N_{\mathrm{err}}}{N_{\mathrm{lig}}}.$$ Every erroneous concealed ligation mitigates the present bias in sequence space and leads to randomness. Erroneous concealed ligation is the reason why the initial sequence patterns decay almost to the random level Z=0.5. However, not all hybridization sites, particularly the shorter ones, have a dehybridization rate determined by the lower bound. As the initial bias for binary zebra motifs on the system level decreases and sequences become more random, non-concealed mismatches in shorter hybridization sites become more likely. Hence, short hybridization sites not yet affected by the lower bound for the unbinding rate become less stable on average and contribute less to the growth process. This explains why the small maxima in the mean length and the other observables shown in Figure 3a,d–f disappear as the bias for binary zebra motifs fades away.

### 3.6. Kinetic Stalling in the Absence of Energetic Bias

We have seen above that concealed erroneous ligations suppress sequence selection during the self-assembly process. However, in the presence of kinetic stalling, concealed erroneous ligations should be reduced. In this section, we thus investigate the (more realistic) model variant, where kinetic stalling is included. We vary the strength of the kinetic stalling factor $\sigma_{1}$ from 0 to 0.1, while fixing $\sigma_{2}=0.1$.

Initially, the dynamics of the mean length $\bar{L}$ are qualitatively similar to the systems without kinetic stalling studied before (see Figure 3a and Figure 4a). However, the onset of growth $\hat{t}$ appears later. The time point of the onset $\hat{t}$ can be predicted by a formula analogous to Equation (Equation 19), which considers the kinetic stalling effect, with an error <15%. For $\sigma_{1}=0.05$, the values for $\hat{t}$ from the prediction and the formal definition are highlighted by the dotted and dashed lines in Figure 4a (for details, see Appendix E and Appendix A). On larger timescales, the model including kinetic stalling deviates from the earlier model. After the steep increase, $\bar{L}$ does not directly settle to a steady-state. Instead, it grows gradually and converges to a constant value eventually. Visualizations with a linear *x*- and a logarithmic or linear *y*-axis reveal that the initial increase after the lag phase is approximately exponential, while the increase during the second growth phase is approximately linear (see Appendix A). For $\sigma_{1}=0,0.05$ or 0.067, similar stationary mean lengths are reached. However, the relaxation time increases with $\sigma_{1}$, such that it takes more than ten times longer for a system with $\sigma_{1}=0.067$ (see inset of Figure 4a) to converge to the stationary state than for a system with infinite stalling. For $\sigma_{1}\leq 0.067$, the steady-state value of the mean length is more than twice as large as for the $\sigma_{1}=\sigma_{2}=1$ scenario. Moreover, the length distributions in the stationary state look qualitatively similar to the ones seen earlier (see Appendix A). For $\sigma_{1}=0.1$, the increase of the mean length during the second growth phase is small during the time window of observation. From Figure 4a, we can not deduce whether the mean length already approached a stationary value, or whether it will keep on growing. If a stationary value was reached, it would be significantly smaller than for $\sigma_{1}=0.67,0.05,0$. For $\sigma_{1}=0.1$ (as well as for $\sigma_{1}=1$), the simulation times are large and prevented us from analyzing at longer time scales. However, plotting the curve for $\sigma_{1}=0.1$ in a coordinate system with a linear *x*-axis might suggest that the system has indeed already converged to a stationary state (see Appendix A). We will discuss the behavior for $\sigma_{1}=0.1$ in more detail in Section 3.9 and provide further evidence why the behavior, in this case, might be qualitatively different from the behavior for $\sigma_{1}=0.67,0.05,0$.

**Figure 4 life-12-00567-f004:**
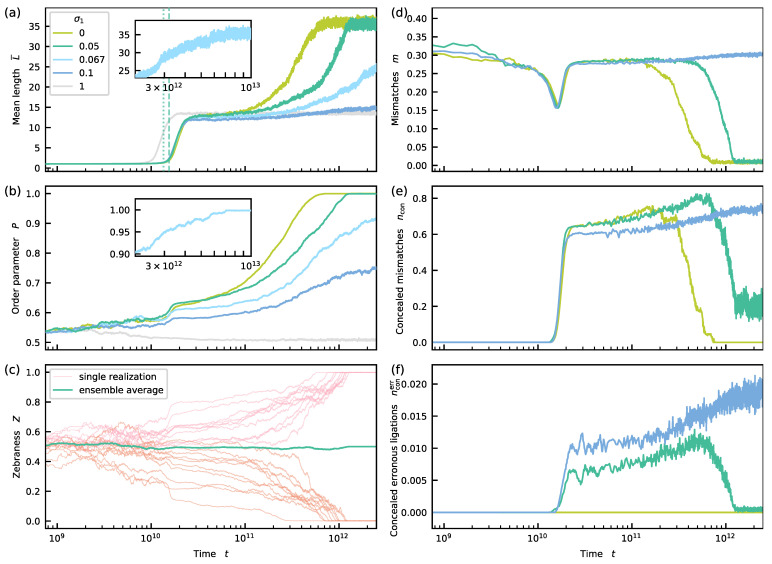
Evolution of mean length $\bar{L}$, sequence order parameter P, and system-level zebraness *Z* in a kinetic stalling scenario without energetic bias, i.e., Δγ=0. For reference, we show the $\sigma_{1}=\sigma_{2}=1$ curves (gray). (**a**) For $\sigma_{1}=\sigma_{2}<1$, $\bar{L}$ shows two distinct growth phases. While the first one is rapid, the second one is slow. Relaxation to the steady-state for $\sigma_{1}=0.067$ appears much later (see inset). For Δγ=0.05, the dashed line corresponds to $\hat{t}$ resulting from the formal definition, whereas the dotted line is the prediction. (**b**) The bias for alternating or homogeneous patterns established in the first growth phase becomes amplified during the second growth phase. For strong stalling ($\sigma_{1}\leq 0.067$), the final pool comprises either pure zebra or fully homogeneous sequences. (**c**) The symmetry of the initial state is broken spontaneously. For $\sigma_{1}=0.05$, equal fractions of realizations evolve to the zebra or homogeneous state; (**c**–**f**) Dynamics of mismatches, concealed mismatches, and concealed erroneous ligations for $\sigma_{1}=0,0.05,0.1$. For details, see the main text.

Is the novel behavior of the mean length shown in Figure 4a related to a novel motif-selective dynamics in sequence space? We now investigate the evolution of the strands’ sequences. Since the initial pool of sequences is symmetric and since neither zebra nor homogeneous binary motifs are preferred energetically, we do not expect a preference for a single realization to go to either a zebra (Z>0.5) or a non-zebra (Z<0.5) state. Hence, the system-level zebraness *Z* is not appropriate to describe an ensemble of realizations. As a meaningful observable to quantify the sequences bias on the ensemble level, we, therefore, introduce the *sequence order parameter* P as
(26)$$P=\max\left\{Z,1-Z\right\}.$$ During the first growth phase of $\bar{L}$, a bias P>0.5 is established for all values of $\sigma_{1}$. The dominance of the bias correlates with the strength of the kinetic stalling. During the slow second growth phase, P gradually increases and reaches a stationary value simultaneously with $\bar{L}$ (see Figure 4b and Appendix A). For $\sigma_{1}=0,0.05$, or 0.067, we observe a value of P≈1 in the stationary state. Hence, on the realization level, the final pool contains either (almost) entirely alternating or homogeneous sequences. We classify such states in sequence space as *pure*. For $\sigma_{1}=0.1$, the final sequence composition within the observation time window is also is non-random. However, the patterns are not pure, i.e., 0.5<P<1. We refer to these states as *partially mixed* states. Whether the system has already converged to a stationary state with P<1 or will further evolve to a pure state as for $\sigma_{1}=0.67,0.05,0$ remains unclear at this point (see above). Figure 4c displays the evolution of the zebraness of all realizations forming the steady-state for $\sigma_{1}=0.05$. On average, one half of the realizations evolves towards the Z=1 state, while the other half evolves towards the Z=0 state. Hence, the symmetry of the initial state is broken spontaneously: either zebra or homogeneous motifs are selected. (See Appendix A for equivalent plots of single realizations for other $\sigma_{1}$ values.)

### 3.7. Hydrolysis and Stalling Boost Sequence Selection

The previous section revealed a coupling between sequence selection and two distinct growth phases in the kinetic stalling scenario. We now interpret and explain this coupling. Here, we consider the cases $\sigma_{1}=0$ and $\sigma_{1}=0.05$, where all trajectories eventually converge to a pure state. The case where $\sigma_{1}=0.1$ is discussed in Section 3.9.

On the level of individual trajectories, fluctuations lead to a small bias in the motif composition even before the mean length starts to grow rapidly (see Figure 4c). This early asymmetry in the distribution of alternating and homogeneous motifs governs the fate of the realization as seen from Figure 4c.

During the first growth phase (see Figure 4a), monomers and short strands self-assemble into longer strands. The first growth phase ends when most of the initial monomers are consumed. At the end of this initial growth phase, a significant bias towards alternating or homogeneous motifs is present. However, a considerable fraction of binary motifs does not yet reflect system-level bias, i.e., differs from the dominant binary motifs (which are either X−Y and Y−X or X−X and Y−Y). Therefore, mismatches in bound strands are still frequent (see Figure 4d). Moreover, since the average strand length is already significantly above $l_{\mathrm{low}}$, most mismatches are concealed (see Figure 4e).

When monomers and short strands do not dominate the pool anymore, hydrolysis becomes important. Every time a strand breaks, an existing binary motif vanishes. During the second growth phase, fragments of broken strands are reassembled to longer strands via ligation on a template strand. (For an analysis of sequence patterns of strands of specific lengths, see Appendix A). If the kinetic stalling is strong, the ligation of two strands is (almost) impossible if a mismatch occurs at the ligation site and the fraction of concealed erroneous ligations is (close to) zero (see Figure 4f). Hence, every ligation forms a new binary motif that (almost) always complements the templating motif at the ligation site. If the templating motif is zebra-like (homogeneous), the new motif is zebra-like (homogeneous) too. Over time, all binary motifs created during the initial growth phase, particularly those that do not reflect the system bias, get destroyed at a uniform rate. At the same time, binary motifs emerging during the second growth phase likely reflect the system bias. Consequently, the bias becomes self-enhancing (see Figure 4b). Newly created binary motifs enhance the system bias even more, while all motifs that do not reflect the asymmetry in motif space become extinct eventually. As a result, the motif composition becomes more and more ordered and mismatches become rarer (see Figure 4d). Remaining mismatches are now even more unlikely to affect the dehybridization rate since the rate is determined by the lower bound $k_{\mathrm{low}}$ for long and primarily complementary hybridization sites. Hence, the fraction of concealed mismatches increases slightly (see Figure 4e). Moreover, if kinetic stalling is finite ($\sigma_{1}>0$), the fraction of concealed erroneous ligations also slightly increases (see Figure 4f).

Since the energetic properties are symmetric, every realization randomly approaches either the zebra-like or homogeneous state. As the pool becomes more complementary in its motif composition, triplex configurations achieve higher stability on average. This increase of stability enhances the probability of templated ligations. Therefore, the symmetry breaking in sequence space is concomitant with enhanced growth, which becomes apparent in the further increase of the mean length. For $\sigma_{1}=0$ and $\sigma_{1}=0.05$, the second growth phase ends when the system has reached an (almost) entirely alternating or homogeneous state (see Figure 4a,b). At that point, the fraction of mismatches is close to zero. The few mismatches that still occur are mostly due to strayed mononucleotides and short oligomers sitting on longer strands. Since these mononucleotides and short oligomers are far from being affected by the lower bound and unbind quickly, the fraction of concealed mismatches takes small values again.

### 3.8. Energetic Bias in the Presence of Kinetic Stalling

The previous section revealed that spontaneous motif selection occurs as a result of kinetic stalling. Without energetic bias, the sequence pool converges to a stationary state which is either dominated by homogenous or alternating sequences. In addition, the energetic symmetry can be broken explicitly if a small energetic bias favoring zebra motifs is applied. To understand the emergent phenomena in this setting, we study two systems, one with strong and one with weak kinetic stalling for various energetic biases Δγ<0.

First, we consider the case where $\sigma_{1}=0.05$. Most parts of the description in the previous section (Δγ=0) also apply here (see Figure 5a,b). Again, we can predict the onset of growth $\hat{t}$ (for details, see Appendix E and Appendix A). The steady-state value of $\bar{L}$ depends weakly on the energetic bias. However, the final state is reached earlier if the bias is stronger. In sequence space, all trajectories end in a pure zebra state Z=1. Hence, symmetry breaking in sequence space is now induced energetically as expected because of the explicit symmetry breaking in the energy landscape.

Second, we investigate a scenario with $\sigma_{1}=\sigma_{2}=0.1$. The mean length grows strongly in the beginning as before (see Figure 5c and, for more details, see Appendix E and Appendix A). The fast growth phase is followed by either a marginal increase (Δγ=−0.1,−0.2) or decrease (Δγ=−0.3) of $\bar{L}$ to a stationary value correlating with the strength of the energetic bias. A strong zebra pattern Z>0.5 is induced in sequence space during the initial increase of the mean length (see Figure 5d). While $\bar{L}$ grows (decays) during the second phase, *Z* also grows (decays). Eventually, the sequence pool converges to a partially mixed stationary state with a significant majority of zebra motifs such that 0.5<Z<1. The excess of zebra motif again correlates with the strength of the energetic bias. Moreover, from Appendix A, it becomes clear that all single trajectories behave similarly to the ensemble mean, i.e., show steady-state values of the zebraness above 0.5.

**Figure 5 life-12-00567-f005:**
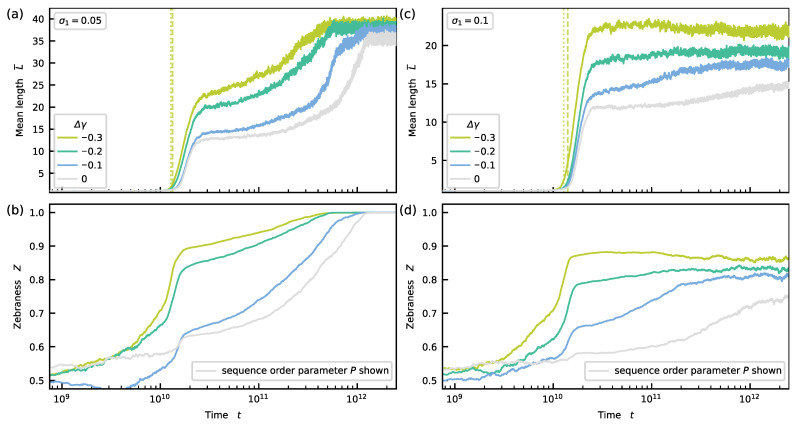
Left column: scenario with $\sigma_{1}=0.05$, right column: scenario with $\sigma_{1}=0.1$ (**a**) The mean length $\bar{L}$ grows again in two steps. The stronger the bias, the earlier the relaxation to the stationary value. For Δγ=−0.3, the dashed line corresponds to $\hat{t}$ resulting from the formal definition, whereas the dotted line is the prediction (same for (**c**)). (**b**) Pronounced zebra patterns emerge during the first growth phase. The patterns become pure during the second growth phase. (**c**) A gradual increase or decay follows the rapid growth phase. The steady-state value of $\bar{L}$ correlates with the strength of the energetic bias. (**d**) While $\bar{L}$ slightly increases (decreases), *Z* also increases (decreases). The sequence pool in the stationary-state shows mixed patterns dominated by zebra motifs. The fraction of zebra motifs depends on the energetic bias. (**a**–**d**) For reference, we also show the sequence order parameter P for the Δγ=0 curves (gray).

### 3.9. Weak versus Strong Kinetic Stalling

In Section 3.6, we speculated whether the system with $\sigma_{1}=0.1$ and without energetic bias (blue curve in Figure 4) reaches a stationary state characterized by P<1 in contrast to the scenarios with $\sigma_{1}=0.067,0.5,0$, where P=1 and referred to the corresponding plot with a linear *x*-axis (see Appendix A). However, this plot did not allow for a clear conclusion either. It could be that the alleged partially mixed stationary state is only transient and that a pure state is reached on much larger time scales. Though the findings from Section 3.8 suggest that the stationary state of the system with $\sigma_{1}=0.1$ and without energetic bias is indeed qualitatively different from the scenarios with $\sigma_{1}\leq 0.067$ and without energetic and characterized by P<1. The curves for $\sigma_{1}=0.1$ and various energetic biases clearly converge to stationary states with P<1 (see colored curves in Figure 5c,d). If the stationary state is partially mixed in the presence of an energetic bias, it is not too far-fetched to assume that it is also partially mixed if the energetic bias is absent.

Naturally, the question arises whether a critical value for $\sigma_{1}$ exists above which the system always reaches a pure state with P=1 for a given value of the energetic bias Δγ. This first question directly leads to a second question, namely, what would be the nature of the corresponding non-equilibrium phase transition? We leave the answer to this question open for future research. However, finding an answer might be challenging since the relaxation time to the stationary state will probably diverge. At this point, we content ourselves with hypothesizing that two different regimes might exist without drawing an exact border: For *strong* kinetic stalling, the system converges to a *pure* state, while, for *weak* kinetic stalling, it converges to a *partially mixed* state.

In the light of the above hypothesis, we now analyze the dynamics of mismatches and concealed erroneous ligations in the energetically unbiased system case for $\sigma_{1}=0.1$ (see the blue curve in Figure 4d–f as for $\sigma_{1}=0$ and $\sigma_{1}=0.05$. The self-amplification of the dominant binary motif comes to a halt during the second growth phase (see Figure 4a Consequently, the fraction of concealed mismatches does not decrease again as for $\sigma_{1}=0$ and $\sigma_{1}=0.05$. Since most of the strands are long enough, most of the mismatches are concealed. Moreover, concealed erroneous ligations are not fully suppressed and still occur in the stationary state. Hence, the weak kinetic stalling scenario includes features of both dynamic regimes described in Section 3.5 and Section 3.7.

## 4. Discussion

### 4.1. Summary

Our study investigated the self-assembly of prebiotic polymers via templated ligation inside a non-equilibrium RNA reactor. We identified kinetic stalling as a critical factor for self-enhanced sequence selection. The final sequence space shows no sequence patterns if the underlying stacking energies are uniform without kinetic stalling. In contrast, spontaneous symmetry breaking occurs for strong kinetic stalling. The final pool contains either entirely homogenous or zebra-like sequences. In scenarios without kinetic stalling, any energetically induced sequence bias vanishes almost completely as strands grow. In contrast, in the presence of kinetic stalling, subtle differences in the stacking energies trigger cascades of self-amplification, leading to highly ordered sequence pools. Our results hint towards the existence of two different stalling regimes. We hypothesize that, for strong kinetic stalling, the system converges to a pure state with P=1, while, for weak kinetic stalling, it converges to a partially mixed state characterized by P<1.

Initially, the mean length shows burst-like growth dynamics after a short lag phase. The onset of the rapid growth coincides with the time point where higher-order ligations become more abundant than ligations joining two monomers and can be predicted analytically.

### 4.2. Prior Work, Our Model, and Future Extensions

In an earlier model for prebiotic self-assembly, strands only grow via random ligation [82]. There, self-folding and complex formation introduced a protection mechanism against hydrolysis for double-stranded segments. Moreover, the ensemble of strands was assumed to reach a binding equilibrium immediately after a random ligation occurred. In this model, protection against hydrolysis could extend the system’s sequence memory. However, the effect was only transient, and all selected patterns vanished eventually. A more recent study combines random ligation and protection against hydrolysis with growth via templated polymerization by mononucleotides [83]. The authors demonstrate that polynucleotides exceeding lengths of 100 can emerge under plausible conditions. Moreover, the authors show that a considerable fraction of the emerging strands forms ribozyme- and tRNA-like secondary structures using folding software. However, the study does not investigate correlations in sequence space on the system level.

Previous theoretical studies considering growth via templated ligation generally explored effective models that reduce the state space to (sub-)sequences without considering complex formation explicitly [33,77,84,85,86,87,88,89,90,91,92]. Such approaches do not treat (de-)hybridization and ligation as elementary steps. Instead, the reactions are coarse-grained into one extension process. The specification of the corresponding rate neglects the intricacies of the assembly mechanisms and requires a priori assumptions regarding the relevant configurations [33,84,85,86,87,90]. Moreover, many models ignore that the hybridization energy is a function of the number and nature of the paired nucleotides and use constant (de-)hybridization rates [33,84,85,86,87,89,90,91]. Other studies treat the sequence dependence of (de-)hybridization employing mean-field approximations where sequence correlations are dismissed [77]. Such simplifications result in systems effectively containing only one type of self-complementary nucleotide [56] and any form of sequence selection is necessarily absent. In contrast, our stochastic approach explicitly takes the sequence-dependent thermodynamic and kinetic aspects of templated ligation into account.

Although being already quite complex, our model also made simplifying assumptions. Future studies need to relax some of our assumptions. In particular, one has to consider nonlinear complexes containing loops and multiple branches. Such configurations can give rise to self-folding, self-templating, template inhibition, and gelation [55,82,93]. All these features potentially influence the sequence dynamics. However, we expect that these effects only become important in the long time limit once the strands have reached sufficient size for the formation of secondary structures. Consequently, the discussion of the emergence of structured sequences on shorter timescales in the limit of strong kinetic stalling is expected to hold, even if secondary structures are taken into account. In addition, one has to extend the alphabet size from two to four. The question here is whether pure states containing only a minimal number of sub motifs exist. In the future, the model could also include additional reactions such as non-templated polymerization and ligation, and recombination [82,89,94,95,96,97,98] or length selective environments [99]. The first two reactions probably play a role in the formation of the first short oligomers, whereas a flow-through system preferentially accumulating long strands can escalate polymerization [100]. Moreover, our study assumed a well-mixed system. Introducing a spatial component together with size-dependent diffusion constants as in Ref. [101], one could study under which conditions local clusters of sequences with specific patterns can emerge and coexist.

### 4.3. Plausibility of a Binary Alphabet

Our study assumed a binary alphabet following previous theoretical work [31,86,87,91,92,102,103,104]. While this assumption simplified the analysis, there is also evidence for a two-letter alphabet preceding the four-letter alphabet [9,10,49,105,106,107]. The plausibility is also underlined by the fact that functional sequences composed of only two types exist [108,109]. For the sake of generality, we referred to the two types of nucleotides appearing in our model as *X* and *Y*. This terminology was motivated by the idea that a pre-RNA, sometimes called *prebioitic XNA*, or alternative RNA nucleotides, may have existed before the modern RNA came into being [110,111,112,113,114,115,116]. Various backbone chemistries [117,118,119,120,121,122], non-canonical nucleotides [105,123,124,125,126,127], and chemical modifications [128,129,130] are eligible, some of which, e.g., PNA and TNA are more plausible to emerge [131,132,133] under the conditions on the early Earth than RNA.

### 4.4. Significance for the Emergence of Life

What is the origin of the first ribozymes heralding the transition from the pre-RNA to the RNA world? In Darwinian evolution, the assembly of low-level building blocks into higher-level entities triggered significant developments [134]. In the light of this evolutionary principle, a multi-step process towards greater complexity, eventually resulting in functionality, also seems natural in prebiotic evolution. Here, we studied one of the first steps following the emergence of early nucleotides. This step forms oligonucleotides displaying distinct sequence patterns that could serve as building blocks for the next higher level of self-organization towards functional ribozymes.

In our study, we considered model variants with and without kinetic stalling. Since kinetic stalling is probably inevitable in non-enzymatic templated ligation [32,38,39,40], the no-stalling variant may appear unwarranted. However, this model variant is essential to separate the effects and identify kinetic stalling as a crucial mechanism enabling self-enhancing sequence selection (see Section 3.5). Moreover, the strength of the stalling effect depends on the underlying activation and nucleotide chemistry [32,38,58] and both weak and strong kinetic stalling scenarios, potentially leading to qualitatively different outcomes, are plausible (see Section 3.6, Section 3.8 and Section 3.9). Furthermore, in a pre-RNA world scenario, a primitive ribozyme catalyzing ligations might have a poor ability to discriminate mismatching ends kinetically. In this case, the ribozyme would operate in a regime where thermodynamics mainly control the discrimination between complementary and non-complementary nucleotides. This regime would be close to the no-stalling model variant.

Moreover, our study revealed that minor differences in the motif-dependent stacking energies significantly affect the dynamics in sequence space. The experiments of Refs. [18,55] probably did not capture this effect, due to the design of the respective experimental systems. These studies used DNA 12- or 20-mers as basic building blocks and a ligase to catalyze bond formation. The ligase requires significant overlaps of both strands with the template to work efficiently. Moreover, the experiments were performed under temperature cycling. The applied temperature cycling was too fast for the long strands to reach a binding equilibrium. Therefore, the hybridization timescale of mostly complementary hybridization sites is always set by the duration of the cold phases. Hence, subtle variations in the stacking energies are not visible. In contrast, the effective cycling in our model is slow enough for thermodynamics to govern the (de-)hybridization of short strands leading to an amplification of the stacking bias (see Section 3.8). The cut-off of the dehybridization rate only affects longer strands emerging from the pool that is already biased.

In non-enzymatic scenarios involving kinetic stalling, the strands formed from the initial pool are already the result of a primary selection process. Selection is not imposed externally but stems from a self-organizing replication network [135]. We showed that the ability to form self-organizing and self-amplifying replication networks is inherent in template-directed growth and does not require higher-level mechanisms such as sequence-specific template inhibition as a result of self-folding, reported previously [55]. In the 1980s, Kauffmann promoted the concept of *autocatalytic sets*—sets of molecules that mutually catalyze their formations [136]—as a chemical intermediate on the way to biological life [137]. Since then, autocatalytic sets have been the subject of many theoretical and experimental studies [29,30,138,139,140,141,142,143]. Once our system has reached a stationary state in the strong kinetic stalling regime, it shows the key features of such an autocatalytic set: strands with a specific pattern promote the formation of new strands of arbitrary length, showing the same pattern. Importantly, this concrete realization of an autocatalytic set emerges naturally from an unstructured initial pool without requiring any external (pre-)engineering. This insight could bridge the gap between strand formation and self-sustaining sequence replication.

## Figures and Tables

**Figure 2 life-12-00567-f002:**
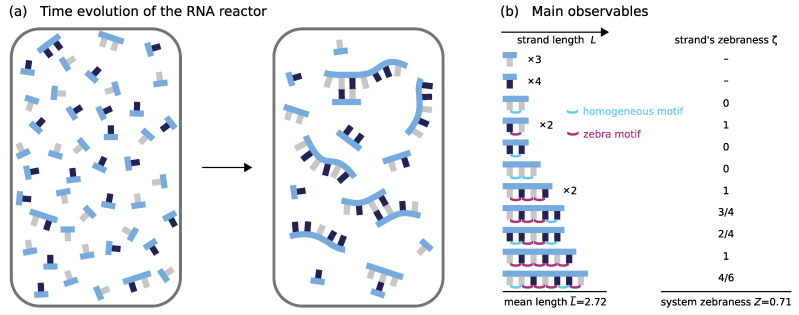
(**a**) Schematic illustration of the time evolution of the non-equilibrium RNA reactor. The reactor is initialized symmetrically with mononucleotides and a few dimers such that the amounts of *X* and *Y* nucleotides are equal and that all four dimer sequences have the same concentrations (see Section 3.1). Within the RNA reactor, oligomers grow via templated ligation and degrade via hydrolysis. Eventually, the sequence pool converges to a non-equilibrium stationary state characterized by its length and sequence distribution (see Figure 1); (**b**) To characterize the dynamics in sequence space, we introduce the zebraness ζ on the level of single strands and the system-level zebraness *Z*. The zebraness ζ of a single strand is the fraction of zebra motifs, i.e., alternating binary motifs contained in the strand. In contrast, the system-level zebraness *Z* measures how zebra-like, i.e., alternating or homogeneous, the pool is as a whole. *Z* corresponds to the total number of zebra motifs spread over all strands normalized with respect to the overall number of binary motifs within all strands present in the reactor.

**Table 1 life-12-00567-t001:** $\bar{\gamma_{\mathrm{com}}}$: mean stacking energies for complementary nearest neighbor blocks in binary RNA and DNA systems at a reference temperature of 37 ${}^{\circ }C$ in units of $k_{B}T$ [36,37]. Δγ: difference between alternating and homogeneous blocks (see Equation (Equation 12)). Note that the sign of Δγ depends on whether A and U or T or G and C are considered for the binary system.

System	RNA		DNA	
Nucleotides	A, U	C, G	A, T	C, G
$\bar{\gamma_{\mathrm{com}}}$	−1.74	−5.00	−1.40	−3.26
Δγ	−0.46	0.60	0.42	−0.65

**Table 2 life-12-00567-t002:** Summary of parameters used in Section 3.

Process	Parameter	Value
hybridization	$k_{\mathrm{coll}}$	1
	$c_{\mathrm{tot}}$	$0.01\;c_{\circ }$
dehybridization	$\bar{\gamma_{\mathrm{com}}},\gamma_{1\mathrm{nc}},\gamma_{2\mathrm{nc}},\Delta \gamma$	$-1.25,0.375,0.75,\left[-0.3,0\right]$
	$l_{\mathrm{low}}$	7
ligation	$l_{\mathrm{lig}}$	10
	$\sigma_{1},\sigma_{2}$	$\left[0,1\right]$, $\left[0.1,1\right]$
hydrolysis	$l_{\mathrm{cut}}$	18.5

## Data Availability

The datasets that support the findings of this study, as well as the computer code used to generate these data, are available from the corresponding author upon reasonable request.

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
