# Peer review of "Thermodynamic and Kinetic Sequence Selection in Enzyme-Free Polymer Self-Assembly inside a Non-equilibrium RNA Reactor"

_life, 2022, doi:10.3390/life12040567_

Round 1
Reviewer 1 Report
My comments are given in the attached file

Reviewer 2 Report
Tobias et al. has studied the self-assembly of RNAs via templated ligation inside a non-equilibrium RNA reactor. They have looked the thermodynamic and kinetic sequence selection by modelling the reactor. I recommend to publish this work in Life.
Questions and comments:
- in line 47, the authors talking about from pre-RNA world to RNA world. It is a bit confused me about the author's definition of pre-RNA world. It looks not like the XNA world in this paper. Can authors rephrase this word to make it more clear?
- in line 58-59, The most contribution of RNA/DNA stability should be the hydrogen bond of base pairs not the base stacking. Ref 35 is not a good ref for authors' argument.
- The hydrolysis of RNA giving 2'3'-cycle phosphate, but the ligation in this paper more likely to talking about the 5'-phosphate ligation (like 2-aminoimidazolide phosphate in Jack's work). Thus, the formation and the hydrolysis of RNA is not an equilibrium. It is two irreversible reactions. The authors should clarify it in line 121.
